# Max-Margin Deep Generative Models

**Chongxuan Li[†], Jun Zhu[†], Tianlin Shi[‡], Bo Zhang[†]**

[†]Dept. of Comp. Sci. & Tech., State Key Lab of Intell. Tech. & Sys., TNList Lab,
Center for Bio-Inspired Computing Research, Tsinghua University, Beijing, 100084, China
[‡]Dept. of Comp. Sci., Stanford University, Stanford, CA 94305, USA
`{licx14@mails., dcszj@, dcszb@}tsinghua.edu.cn; stl501@gmail.com`

## Abstract

Deep generative models (DGMs) are effective on learning multilayered representations of complex data and performing inference of input data by exploring the generative ability. However, little work has been done on examining or empowering the discriminative ability of DGMs on making accurate predictions. This paper presents max-margin deep generative models (mmDGMs), which explore the strongly discriminative principle of max-margin learning to improve the discriminative power of DGMs, while retaining the generative capability. We develop an efficient doubly stochastic subgradient algorithm for the piecewise linear objective. Empirical results on MNIST and SVHN datasets demonstrate that (1) max-margin learning can significantly improve the prediction performance of DGMs and meanwhile retain the generative ability; and (2) mmDGMs are competitive to the state-of-the-art fully discriminative networks by employing deep convolutional neural networks (CNNs) as both recognition and generative models.

## 1 Introduction

Max-margin learning has been effective on learning discriminative models, with many examples such as univariate-output support vector machines (SVMs) [5] and multivariate-output max-margin Markov networks (or structured SVMs) [30, 1, 31]. However, the ever-increasing size of complex data makes it hard to construct such a fully discriminative model, which has only single layer of adjustable weights, due to the facts that: (1) the manually constructed features may not well capture the underlying high-order statistics; and (2) a fully discriminative approach cannot reconstruct the input data when noise or missing values are present.

To address the first challenge, previous work has considered incorporating latent variables into a max-margin model, including partially observed maximum entropy discrimination Markov networks [37], structured latent SVMs [32] and max-margin min-entropy models [20]. All this work has primarily focused on a shallow structure of latent variables. To improve the flexibility, learning SVMs with a deep latent structure has been presented in [29]. However, these methods do not address the second challenge, which requires a generative model to describe the inputs. The recent work on learning max-margin generative models includes max-margin Harmoniums [4], max-margin topic models [34, 35], and nonparametric Bayesian latent SVMs [36] which can infer the dimension of latent features from data. However, these methods only consider the shallow structure of latent variables, which may not be flexible enough to describe complex data.

Much work has been done on learning generative models with a deep structure of nonlinear hidden variables, including deep belief networks [25, 16, 23], autoregressive models [13, 9], and stochastic variations of neural networks [3]. For such models, inference is a challenging problem, but fortunately there exists much recent progress on stochastic variational inference algorithms [12, 24]. However, the primary focus of deep generative models (DGMs) has been on unsupervised learning,

with the goals of learning latent representations and generating input samples. Though the latent representations can be used with a downstream classifier to make predictions, it is often beneficial to learn a joint model that considers both input and response variables. One recent attempt is the conditional generative models [11], which treat labels as conditions of a DGM to describe input data. This conditional DGM is learned in a semi-supervised setting, which is not exclusive to ours.

In this paper, we revisit the max-margin principle and present a max-margin deep generative model (mmDGM), which learns multi-layer representations that are good for both classification and input inference. Our mmDGM conjoins the flexibility of DGMs on describing input data and the strong discriminative ability of max-margin learning on making accurate predictions. We formulate mmDGM as solving a variational inference problem of a DGM regularized by a set of max-margin posterior constraints, which bias the model to learn representations that are good for prediction. We define the max-margin posterior constraints as a linear functional of the target variational distribution of the latent presentations. Then, we develop a doubly stochastic subgradient descent algorithm, which generalizes the Pagesos algorithm [28] to consider nontrivial latent variables. For the variational distribution, we build a recognition model to capture the nonlinearity, similar as in [12, 24].

We consider two types of networks used as our recognition and generative models: multiple layer perceptrons (MLPs) as in [12, 24] and convolutional neural networks (CNNs) [14]. Though CNNs have shown promising results in various domains, especially for image classification, little work has been done to take advantage of CNN to generate images. The recent work [6] presents a type of CNN to map manual features including class labels to RBG chair images by applying unpooling, convolution and rectification sequentially; but it is a deterministic mapping and there is no random generation. Generative Adversarial Nets [7] employs a single such layer together with MLPs in a minimax two-player game framework with primary goal of generating images. We propose to stack this structure to form a highly non-trivial deep generative network to generate images from latent variables learned automatically by a recognition model using standard CNN. We present the detailed network structures in experiments part. Empirical results on MNIST [14] and SVHN [22] datasets demonstrate that mmDGM can significantly improve the prediction performance, which is competitive to the state-of-the-art methods [33, 17, 8, 15], while retaining the capability of generating input samples and completing their missing values.

## 2 Basics of Deep Generative Models

We start from a general setting, where we have $N$ i.i.d. data $\mathbf{X} = \{\mathbf{x}_n\}_{n=1}^N$. A deep generative model (DGM) assumes that each $\mathbf{x}_n \in \mathbb{R}^D$ is generated from a vector of latent variables $\mathbf{z}_n \in \mathbb{R}^K$, which itself follows some distribution. The joint probability of a DGM is as follows:

$$p(\mathbf{X}, \mathbf{Z}|\boldsymbol{\alpha}, \boldsymbol{\beta}) = \prod_{n=1}^N p(\mathbf{z}_n|\boldsymbol{\alpha})p(\mathbf{x}_n|\mathbf{z}_n, \boldsymbol{\beta}), \tag{1}$$

where $p(\mathbf{z}_n|\boldsymbol{\alpha})$ is the prior of the latent variables and $p(\mathbf{x}_n|\mathbf{z}_n, \boldsymbol{\beta})$ is the likelihood model for generating observations. For notation simplicity, we define $\boldsymbol{\theta} = (\boldsymbol{\alpha}, \boldsymbol{\beta})$. Depending on the structure of $\mathbf{z}$, various DGMs have been developed, such as the deep belief networks [25, 16], deep sigmoid networks [21], deep latent Gaussian models [24], and deep autoregressive models [9]. In this paper, we focus on the directed DGMs, which can be easily sampled from via an ancestral sampler.

However, in most cases learning DGMs is challenging due to the intractability of posterior inference. The state-of-the-art methods resort to stochastic variational methods under the maximum likelihood estimation (MLE) framework, $\hat{\boldsymbol{\theta}} = \operatorname{argmax}_{\boldsymbol{\theta}} \log p(\mathbf{X}|\boldsymbol{\theta})$. Specifically, let $q(\mathbf{Z})$ be the variational distribution that approximates the true posterior $p(\mathbf{Z}|\mathbf{X}, \boldsymbol{\theta})$. A variational upper bound of the per sample negative log-likelihood (NLL) $-\log p(\mathbf{x}_n|\boldsymbol{\alpha}, \boldsymbol{\beta})$ is:

$$\mathcal{L}(\boldsymbol{\theta}, q(\mathbf{z}_n); \mathbf{x}_n) \triangleq \mathbf{KL}(q(\mathbf{z}_n)||p(\mathbf{z}_n|\boldsymbol{\alpha})) - \mathbb{E}_{q(\mathbf{z}_n)}[\log p(\mathbf{x}_n|\mathbf{z}_n, \boldsymbol{\beta})], \tag{2}$$

where $\mathbf{KL}(q||p)$ is the Kullback-Leibler (KL) divergence between distributions $q$ and $p$. Then, $\mathcal{L}(\boldsymbol{\theta}, q(\mathbf{Z}); \mathbf{X}) \triangleq \sum_n \mathcal{L}(\boldsymbol{\theta}, q(\mathbf{z}_n); \mathbf{x}_n)$ upper bounds the full negative log-likelihood $-\log p(\mathbf{X}|\boldsymbol{\theta})$.

It is important to notice that if we do not make restricting assumption on the variational distribution $q$, the lower bound is tight by simply setting $q(\mathbf{Z}) = p(\mathbf{Z}|\mathbf{X}, \boldsymbol{\theta})$. That is, the MLE is equivalent to solving the variational problem: $\min_{\boldsymbol{\theta}, q(\mathbf{Z})} \mathcal{L}(\boldsymbol{\theta}, q(\mathbf{Z}); \mathbf{X})$. However, since the true posterior is intractable except a handful of special cases, we must resort to approximation methods. One common

assumption is that the variational distribution is of some parametric form, $q_\phi(\mathbf{Z})$, and then we optimize the variational bound w.r.t the variational parameters $\phi$. For DGMs, another challenge arises that the variational bound is often intractable to compute analytically. To address this challenge, the early work further bounds the intractable parts with tractable ones by introducing more variational parameters [26]. However, this technique increases the gap between the bound being optimized and the log-likelihood, potentially resulting in poorer estimates. Much recent progress [12, 24, 21] has been made on hybrid Monte Carlo and variational methods, which approximates the intractable expectations and their gradients over the parameters $(\boldsymbol{\theta}, \phi)$ via some unbiased Monte Carlo estimates. Furthermore, to handle large-scale datasets, stochastic optimization of the variational objective can be used with a suitable learning rate annealing scheme. It is important to notice that variance reduction is a key part of these methods in order to have fast and stable convergence.

Most work on directed DGMs has been focusing on the generative capability on inferring the observations, such as filling in missing values [12, 24, 21], while little work has been done on investigating the predictive power, except the semi-supervised DGMs [11] which builds a DGM conditioned on the class labels and learns the parameters via MLE. Below, we present max-margin deep generative models, which explore the discriminative max-margin principle to improve the predictive ability of the latent representations, while retaining the generative capability.

## 3    Max-margin Deep Generative Models

We consider supervised learning, where the training data is a pair $(\mathbf{x}, y)$ with input features $\mathbf{x} \in \mathbb{R}^D$ and the ground truth label $y$. Without loss of generality, we consider the multi-class classification, where $y \in \mathcal{C} = \{1, \ldots, M\}$. A max-margin deep generative model (mmDGM) consists of two components: (1) a deep generative model to describe input features; and (2) a max-margin classifier to consider supervision. For the generative model, we can in theory adopt any DGM that defines a joint distribution over $(\mathbf{X}, \mathbf{Z})$ as in Eq. (1). For the max-margin classifier, instead of fitting the input features into a conventional SVM, we define the linear classifier on the latent representations, whose learning will be regularized by the supervision signal as we shall see. Specifically, if the latent representation $\mathbf{z}$ is given, we define the latent discriminant function $F(y, \mathbf{z}, \boldsymbol{\eta}; \mathbf{x}) = \boldsymbol{\eta}^\top \mathbf{f}(y, \mathbf{z})$, where $\mathbf{f}(y, \mathbf{z})$ is an $MK$-dimensional vector that concatenates $M$ subvectors, with the $y$th being $\mathbf{z}$ and all others being zero, and $\boldsymbol{\eta}$ is the corresponding weight vector.

We consider the case that $\boldsymbol{\eta}$ is a random vector, following some prior distribution $p_0(\boldsymbol{\eta})$. Then our goal is to infer the posterior distribution $p(\boldsymbol{\eta}, \mathbf{Z}|\mathbf{X}, \mathbf{Y})$, which is typically approximated by a variational distribution $q(\boldsymbol{\eta}, \mathbf{Z})$ for computational tractability. Notice that this posterior is different from the one in the vanilla DGM. We expect that the supervision information will bias the learned representations to be more powerful on predicting the labels at testing. To account for the uncertainty of $(\boldsymbol{\eta}, \mathbf{Z})$, we take the expectation and define the discriminant function $F(y; \mathbf{x}) = \mathbb{E}_q\left[\boldsymbol{\eta}^\top \mathbf{f}(y, \mathbf{z})\right]$, and the final prediction rule that maps inputs to outputs is:

$$\hat{y} = \operatorname*{argmax}_{y \in \mathcal{C}} F(y; \mathbf{x}). \tag{3}$$

Note that different from the conditional DGM [11], which puts the class labels upstream, the above classifier is a downstream model, in the sense that the supervision signal is determined by conditioning on the latent representations.

### 3.1    The Learning Problem

We want to jointly learn the parameters $\boldsymbol{\theta}$ and infer the posterior distribution $q(\boldsymbol{\eta}, \mathbf{Z})$. Based on the equivalent variational formulation of MLE, we define the joint learning problem as solving:

$$\min_{\boldsymbol{\theta}, q(\boldsymbol{\eta}, \mathbf{Z}), \boldsymbol{\xi}} \mathcal{L}(\boldsymbol{\theta}, q(\boldsymbol{\eta}, \mathbf{Z}); \mathbf{X}) + C \sum_{n=1}^{N} \xi_n \tag{4}$$

$$\forall n, y \in \mathcal{C}, \text{s.t.} : \begin{cases} \mathbb{E}_q[\boldsymbol{\eta}^\top \Delta \mathbf{f}_n(y)] \geq \Delta l_n(y) - \xi_n \\ \xi_n \geq 0, \end{cases}$$

where $\Delta \mathbf{f}_n(y) = \mathbf{f}(y_n, \mathbf{z}_n) - \mathbf{f}(y, \mathbf{z}_n)$ is the difference of the feature vectors; $\Delta l_n(y)$ is the loss function that measures the cost to predict $y$ if the true label is $y_n$; and $C$ is a nonnegative regularization parameter balancing the two components. In the objective, the variational bound is defined

as $\mathcal{L}(\boldsymbol{\theta}, q(\boldsymbol{\eta}, \mathbf{Z}); \mathbf{X}) = \mathbf{KL}(q(\boldsymbol{\eta}, \mathbf{Z})||p_0(\boldsymbol{\eta}, \mathbf{Z}|\boldsymbol{\alpha})) - \mathbb{E}_q[\log p(\mathbf{X}|\mathbf{Z}, \boldsymbol{\beta})]$, and the margin constraints are from the classifier (3). If we ignore the constraints (e.g., setting $C$ at 0), the solution of $q(\boldsymbol{\eta}, \mathbf{Z})$ will be exactly the Bayesian posterior, and the problem is equivalent to do MLE for $\boldsymbol{\theta}$.

By absorbing the slack variables, we can rewrite the problem in an unconstrained form:

$$\min_{\boldsymbol{\theta}, q(\boldsymbol{\eta}, \mathbf{Z})} \mathcal{L}(\boldsymbol{\theta}, q(\boldsymbol{\eta}, \mathbf{Z}); \mathbf{X}) + C\mathcal{R}(q(\boldsymbol{\eta}, \mathbf{Z}; \mathbf{X})), \tag{5}$$

where the hinge loss is: $\mathcal{R}(q(\boldsymbol{\eta}, \mathbf{Z}); \mathbf{X}) = \sum_{n=1}^{N} \max_{y \in \mathcal{C}}(\Delta l_n(y) - \mathbb{E}_q[\boldsymbol{\eta}^\top \Delta \mathbf{f}_n(y)])$. Due to the convexity of $\max$ function, it is easy to verify that the hinge loss is an upper bound of the training error of classifier (3), that is, $\mathcal{R}(q(\boldsymbol{\eta}, \mathbf{Z}); \mathbf{X}) \geq \sum_n \Delta l_n(\hat{y}_n)$. Furthermore, the hinge loss is a convex functional over the variational distribution because of the linearity of the expectation operator. These properties render the hinge loss as a good surrogate to optimize over. Previous work has explored this idea to learn discriminative topic models [34], but with a restriction on the shallow structure of hidden variables. Our work presents a significant extension to learn deep generative models, which pose new challenges on the learning and inference.

### 3.2 The Doubly Stochastic Subgradient Algorithm

The variational formulation of problem (5) naturally suggests that we can develop a variational algorithm to address the intractability of the true posterior. We now present a new algorithm to solve problem (5). Our method is a doubly stochastic generalization of the Pegasos (i.e., Primal Estimated sub-GrAdient SOlver for SVM) algorithm [28] for the classic SVMs with fully observed input features, with the new extension of dealing with a highly nontrivial structure of latent variables.

First, we make the structured mean-field (SMF) assumption that $q(\boldsymbol{\eta}, \mathbf{Z}) = q(\boldsymbol{\eta})q_{\boldsymbol{\phi}}(\mathbf{Z})$. Under the assumption, we have the discriminant function as $\mathbb{E}_q[\boldsymbol{\eta}^\top \Delta \mathbf{f}_n(y)] = \mathbb{E}_{q(\boldsymbol{\eta})}[\boldsymbol{\eta}^\top]\mathbb{E}_{q_{\boldsymbol{\phi}}(\mathbf{z}^{(n)})}[\Delta \mathbf{f}_n(y)]$. Moreover, we can solve for the optimal solution of $q(\boldsymbol{\eta})$ in some analytical form. In fact, by the calculus of variations, we can show that given the other parts the solution is $q(\boldsymbol{\eta}) \propto p_0(\boldsymbol{\eta}) \exp\left(\boldsymbol{\eta}^\top \sum_{n,y} \omega_n^y \mathbb{E}_{q_{\boldsymbol{\phi}}}[\Delta \mathbf{f}_n(y)]\right)$, where $\boldsymbol{\omega}$ are the Lagrange multipliers (See [34] for details). If the prior is normal, $p_0(\boldsymbol{\eta}) = \mathcal{N}(\mathbf{0}, \sigma^2 \mathbf{I})$, we have the normal posterior: $q(\boldsymbol{\eta}) = \mathcal{N}(\boldsymbol{\lambda}, \sigma^2 \mathbf{I})$, where $\boldsymbol{\lambda} = \sigma^2 \sum_{n,y} \omega_n^y \mathbb{E}_{q_{\boldsymbol{\phi}}}[\Delta \mathbf{f}_n(y)]$. Therefore, even though we did not make a parametric form assumption of $q(\boldsymbol{\eta})$, the above results show that the optimal posterior distribution of $\boldsymbol{\eta}$ is Gaussian. Since we only use the expectation in the optimization problem and in prediction, we can directly solve for the mean parameter $\boldsymbol{\lambda}$ instead of $q(\boldsymbol{\eta})$. Further, in this case we can verify that $\mathbf{KL}(q(\boldsymbol{\eta})||p_0(\boldsymbol{\eta})) = \frac{||\boldsymbol{\lambda}||^2}{2\sigma^2}$ and then the equivalent objective function in terms of $\boldsymbol{\lambda}$ can be written as:

$$\min_{\boldsymbol{\theta}, \boldsymbol{\phi}, \boldsymbol{\lambda}} \mathcal{L}(\boldsymbol{\theta}, \boldsymbol{\phi}; \mathbf{X}) + \frac{||\boldsymbol{\lambda}||^2}{2\sigma^2} + C\mathcal{R}(\boldsymbol{\lambda}, \boldsymbol{\phi}; \mathbf{X}), \tag{6}$$

where $\mathcal{R}(\boldsymbol{\lambda}, \boldsymbol{\phi}; \mathbf{X}) = \sum_{n=1}^{N} \ell(\boldsymbol{\lambda}, \boldsymbol{\phi}; \mathbf{x}_n)$ is the total hinge loss, and the per-sample hinge-loss is $\ell(\boldsymbol{\lambda}, \boldsymbol{\phi}; \mathbf{x}_n) = \max_{y \in \mathcal{C}}(\Delta l_n(y) - \boldsymbol{\lambda}^\top \mathbb{E}_{q_{\boldsymbol{\phi}}}[\Delta \mathbf{f}_n(y)])$. Below, we present a doubly stochastic subgradient descent algorithm to solve this problem.

The *first stochasticity* arises from a stochastic estimate of the objective by random mini-batches. Specifically, the batch learning needs to scan the full dataset to compute subgradients, which is often too expensive to deal with large-scale datasets. One effective technique is to do stochastic subgradient descent [28], where at each iteration we randomly draw a mini-batch of the training data and then do the variational updates over the small mini-batch. Formally, given a mini batch of size $m$, we get an unbiased estimate of the objective:

$$\tilde{\mathcal{L}}_m := \frac{N}{m} \sum_{n=1}^{m} \mathcal{L}(\boldsymbol{\theta}, \boldsymbol{\phi}; \mathbf{x}_n) + \frac{||\boldsymbol{\lambda}||^2}{2\sigma^2} + \frac{NC}{m} \sum_{n=1}^{m} \ell(\boldsymbol{\lambda}, \boldsymbol{\phi}; \mathbf{x}_n).$$

The *second stochasticity* arises from a stochastic estimate of the per-sample variational bound and its subgradient, whose intractability calls for another Monte Carlo estimator. Formally, let $\mathbf{z}_n^l \sim q_{\boldsymbol{\phi}}(\mathbf{z}|\mathbf{x}_n, y_n)$ be a set of samples from the variational distribution, where we explicitly put the conditions. Then, an estimate of the per-sample variational bound and the per-sample hinge-loss is

$$\tilde{\mathcal{L}}(\boldsymbol{\theta}, \boldsymbol{\phi}; \mathbf{x}_n) = \frac{1}{L}\sum_l \log p(\mathbf{x}_n, \mathbf{z}_n^l|\boldsymbol{\beta}) - \log q_{\boldsymbol{\phi}}(\mathbf{z}_n^l); \quad \tilde{\ell}(\boldsymbol{\lambda}, \boldsymbol{\phi}; \mathbf{x}_n) = \max_y \left(\Delta l_n(y) - \frac{1}{L}\sum_l \boldsymbol{\lambda}^\top \Delta \mathbf{f}_n(y, \mathbf{z}_n^l)\right),$$

where $\Delta\mathbf{f}_n(y, \mathbf{z}_n^l) = \mathbf{f}(y_n, \mathbf{z}_n^l) - \mathbf{f}(y, \mathbf{z}_n^l)$. Note that $\tilde{\mathcal{L}}$ is an unbiased estimate of $\mathcal{L}$, while $\tilde{\ell}$ is a biased estimate of $\ell$. Nevertheless, we can still show that $\tilde{\ell}$ is an upper bound estimate of $\ell$ under expectation. Furthermore, this biasedness does not affect our estimate of the gradient. In fact, by using the equality $\nabla_\phi q_\phi(\mathbf{z}) = q_\phi(\mathbf{z}) \nabla_\phi \log q_\phi(\mathbf{z})$, we can construct an unbiased Monte Carlo estimate of $\nabla_\phi(\mathcal{L}(\boldsymbol{\theta}, \phi; \mathbf{x}_n) + \ell(\boldsymbol{\lambda}, \phi; \mathbf{x}_n))$ as:

$$\mathbf{g}_\phi = \frac{1}{L} \sum_{l=1}^{L} \Big( \log p(\mathbf{z}_n^l, \mathbf{x}_n) - \log q_\phi(\mathbf{z}_n^l) + C\boldsymbol{\lambda}^\top \Delta\mathbf{f}_n(\tilde{y}_n, \mathbf{z}_n^l) \Big) \nabla_\phi \log q_\phi(\mathbf{z}_n^l), \qquad (7)$$

where the last term roots from the hinge loss with the loss-augmented prediction $\tilde{y}_n = \operatorname{argmax}_y(\Delta l_n(y) + \frac{1}{L}\sum_l \boldsymbol{\lambda}^\top \mathbf{f}(y, \mathbf{z}_n^l))$. For $\boldsymbol{\theta}$ and $\boldsymbol{\lambda}$, the estimates of the gradient $\nabla_{\boldsymbol{\theta}} \mathcal{L}(\boldsymbol{\theta}, \phi; \mathbf{x}_n)$ and the subgradient $\nabla_{\boldsymbol{\lambda}} \ell(\boldsymbol{\lambda}, \phi; \mathbf{x}_n)$ are easier, which are:

$$\mathbf{g}_{\boldsymbol{\theta}} = \frac{1}{L} \sum_l \nabla_{\boldsymbol{\theta}} \log p(\mathbf{x}_n, \mathbf{z}_n^l | \boldsymbol{\theta}), \quad \mathbf{g}_{\boldsymbol{\lambda}} = \frac{1}{L} \sum_l \big( \mathbf{f}(\tilde{y}_n, \mathbf{z}_n^l) - \mathbf{f}(y_n, \mathbf{z}_n^l) \big).$$

Notice that the sampling and the gradient $\nabla_\phi \log q_\phi(\mathbf{z}_n^l)$ only depend on the variational distribution, not the underlying model.

The above estimates consider the general case where the variational bound is intractable. In some cases, we can compute the KL-divergence term analytically, e.g., when the prior and the variational distribution are both Gaussian. In such cases, we only need to estimate the rest intractable part by sampling, which often reduces the variance [12]. Similarly, we could use the expectation

---

**Algorithm 1** Doubly Stochastic Subgradient Algorithm

Initialize $\boldsymbol{\theta}$, $\boldsymbol{\lambda}$, and $\phi$
**repeat**
    draw a random mini-batch of $m$ data points
    draw random samples from noise distribution $p(\boldsymbol{\epsilon})$
    compute subgradient $\mathbf{g} = \nabla_{\boldsymbol{\theta}, \boldsymbol{\lambda}, \phi} \tilde{\mathcal{L}}(\boldsymbol{\theta}, \boldsymbol{\lambda}, \phi; \mathbf{X}^m, \boldsymbol{\epsilon})$
    update parameters $(\boldsymbol{\theta}, \boldsymbol{\lambda}, \phi)$ using subgradient $\mathbf{g}$.
**until** Converge
**return** $\boldsymbol{\theta}$, $\boldsymbol{\lambda}$, and $\phi$

---

of the features directly, if it can be computed analytically, in the computation of subgradients (e.g., $\mathbf{g}_{\boldsymbol{\theta}}$ and $\mathbf{g}_{\boldsymbol{\lambda}}$) instead of sampling, which again can lead to variance reduction.

With the above estimates of subgradients, we can use stochastic optimization methods such as SGD [28] and AdaM [10] to update the parameters, as outlined in Alg. 1. Overall, our algorithm is a doubly stochastic generalization of Pegasos to deal with the highly nontrivial latent variables.

Now, the remaining question is how to define an appropriate variational distribution $q_\phi(\mathbf{z})$ to obtain a robust estimate of the subgradients as well as the objective. Two types of methods have been developed for unsupervised DGMs, namely, variance reduction [21] and auto-encoding variational Bayes (AVB) [12]. Though both methods can be used for our models, we focus on the AVB approach. For continuous variables $\mathbf{Z}$, under certain mild conditions we can reparameterize the variational distribution $q_\phi(\mathbf{z})$ using some simple variables $\boldsymbol{\epsilon}$. Specifically, we can draw samples $\boldsymbol{\epsilon}$ from some simple distribution $p(\boldsymbol{\epsilon})$ and do the transformation $\mathbf{z} = \mathbf{g}_\phi(\boldsymbol{\epsilon}, \mathbf{x}, y)$ to get the sample of the distribution $q(\mathbf{z}|\mathbf{x}, y)$. We refer the readers to [12] for more details. In our experiments, we consider the special Gaussian case, where we assume that the variational distribution is a multivariate Gaussian with a diagonal covariance matrix:

$$q_\phi(\mathbf{z}|\mathbf{x}, y) = \mathcal{N}(\boldsymbol{\mu}(\mathbf{x}, y; \phi), \boldsymbol{\sigma}^2(\mathbf{x}, y; \phi)), \qquad (8)$$

whose mean and variance are functions of the input data. This defines our recognition model. Then, the reparameterization trick is as follows: we first draw standard normal variables $\boldsymbol{\epsilon}^l \sim \mathcal{N}(0, \mathbf{I})$ and then do the transformation $\mathbf{z}_n^l = \boldsymbol{\mu}(\mathbf{x}_n, y_n; \phi) + \boldsymbol{\sigma}(\mathbf{x}_n, y_n; \phi) \odot \boldsymbol{\epsilon}^l$ to get a sample. For simplicity, we assume that both the mean and variance are function of $\mathbf{x}$ only. However, it is worth to emphasize that although the recognition model is unsupervised, the parameters $\phi$ are learned in a supervised manner because the subgradient (7) depends on the hinge loss. Further details of the experimental settings are presented in Sec. 4.1.

## 4 Experiments

We now present experimental results on the widely adopted MNIST [14] and SVHN [22] datasets. Though mmDGMs are applicable to any DGMs that define a joint distribution of $\mathbf{X}$ and $\mathbf{Z}$, we

concentrate on the Variational Auto-encoder (VA) [12], which is unsupervised. We denote our mmDGM with VA by MMVA. In our experiments, we consider two types of recognition models: multiple layer perceptrons (MLPs) and convolutional neural networks (CNNs). We implement all experiments based on Theano [2]. [1]

## 4.1 Architectures and Settings

In the MLP case, we follow the settings in [11] to compare both generative and discriminative capacity of VA and MMVA. In the CNN case, we use standard convolutional nets [14] with convolution and max-pooling operation as the recognition model to obtain more competitive classification results. For the generative model, we use unconvnets [6] with a "symmetric" structure as the recognition model, to reconstruct the input images approximately. More specifically, the top-down generative model has the same structure as the bottom-up recognition model but replacing max-pooling with unpooling operation [6] and applies unpooling, convolution and rectification in order. The total number of parameters in the convolutional network is comparable with previous work [8, 17, 15]. For simplicity, we do not involve mlpconv layers [17, 15] and contrast normalization layers in our recognition model, but they are not exclusive to our model. We illustrate details of the network architectures in appendix A.

In both settings, the mean and variance of the latent $z$ are transformed from the last layer of the recognition model through a linear operation. It should be noticed that we could use not only the expectation of $z$ but also the activation of any layer in the recognition model as features. The only theoretical difference is from where we add a hinge loss regularization to the gradient and back-propagate it to previous layers. In all of the experiments, the mean of $z$ has the same nonlinearity but typically much lower dimension than the activation of the last layer in the recognition model, and hence often leads to a worse performance. In the MLP case, we concatenate the activations of 2 layers as the features used in the supervised tasks. In the CNN case, we use the activations of the last layer as the features. We use AdaM [10] to optimize parameters in all of the models. Although it is an adaptive gradient-based optimization method, we decay the global learning rate by factor three periodically after sufficient number of epochs to ensure a stable convergence.

We denote our mmDGM with MLPs by **MMVA**. To perform classification using VA, we first learn the feature representations by VA, and then build a linear SVM classifier on these features using the Pegasos stochastic subgradient algorithm [28]. This baseline will be denoted by **VA+Pegasos**. The corresponding models with CNNs are denoted by **CMMVA** and **CVA+Pegasos** respectively.

## 4.2 Results on the MNIST dataset

We present both the prediction performance and the results on generating samples of MMVA and VA+Pegasos with both kinds of recognition models on the MNIST [14] dataset, which consists of images of 10 different classes (0 to 9) of size $28 \times 28$ with 50,000 training samples, 10,000 validating samples and 10,000 testing samples.

Table 1: Error rates (%) on MNIST dataset.

| MODEL | ERROR RATE |
|---|---|
| *VA+Pegasos* | 1.04 |
| *VA+Class-conditionVA* | 0.96 |
| *MMVA* | 0.90 |
| *CVA+Pegasos* | 1.35 |
| *CMMVA* | 0.45 |
| *Stochastic Pooling* [33] | 0.47 |
| *Network in Network* [17] | 0.47 |
| *Maxout Network* [8] | 0.45 |
| *DSN* [15] | 0.39 |

### 4.2.1 Predictive Performance

In the MLP case, we only use 50,000 training data, and the parameters for classification are optimized according to the validation set. We choose $C = 15$ for MMVA and initialize it with an unsupervised pre-training procedure in classification. First three rows in Table 1 compare VA+Pegasos, VA+Class-condtionVA and MMVA, where VA+Class-condtionVA refers to the best fully supervised model in [11]. Our model outperforms the baseline significantly. We further use the t-SNE algorithm [19] to embed the features learned by VA and MMVA on 2D plane, which again demonstrates the stronger discriminative ability of MMVA (See Appendix B for details).

In the CNN case, we use 60,000 training data. Table 2 shows the effect of $C$ on classification error rate and variational lower bound. Typically, as $C$ gets lager, CMMVA learns more discriminative features and leads to a worse estimation of data likelihood. However, if $C$ is too small, the supervision is not enough to lead to predictive features. Nevertheless, $C = 10^3$ is quite a good trade-off

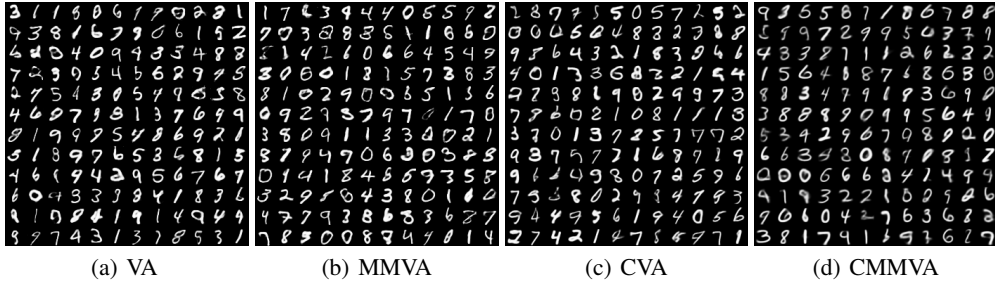

| (a) VA | (b) MMVA | (c) CVA | (d) CMMVA |

Figure 1: (a-b): randomly generated images by VA and MMVA, 3000 epochs; (c-d): randomly generated images by CVA and CMMVA, 600 epochs.

between the classification performance and generative performance and this is the default setting of CMMVA on MNIST throughout this paper. In this setting, the classification performance of our CMMVA model is comparable to the recent state-of-the-art fully discriminative networks (without data augmentation), shown in the last four rows of Table 1.

#### 4.2.2 Generative Performance

We further investigate the generative capability of MMVA on generating samples. Fig. 1 illustrates the images randomly sampled from VA and MMVA models where we output the expectation of the gray value at each pixel to get a smooth visualization. We do not pre-train our model in all settings when generating data to prove that MMVA (CMMVA) remains the generative capability of DGMs.

Table 2: Effects of $C$ on MNIST dataset with a CNN recognition model.

| C | ERROR RATE (%) | LOWER BOUND |
|---|---|---|
| 0 | 1.35 | -93.17 |
| 1 | 1.86 | -95.86 |
| 10 | 0.88 | -95.90 |
| $10^2$ | 0.54 | -96.35 |
| $10^3$ | 0.45 | -99.62 |
| $10^4$ | 0.43 | -112.12 |

### 4.3 Results on the SVHN (Street View House Numbers) dataset

SVHN [22] is a large dataset consisting of color images of size $32 \times 32$. The task is to recognize center digits in natural scene images, which is significantly harder than classification of hand-written digits. We follow the work [27, 8] to split the dataset into 598,388 training data, 6000 validating data and 26, 032 testing data and preprocess the data by Local Contrast Normalization (LCN).

We only consider the CNN recognition model here. The network structure is similar to that in MNIST. We set $C = 10^4$ for our CMMVA model on SVHN by default.

Table 3 shows the predictive performance. In this more challenging problem, we observe a larger improvement by CMMVA as compared to CVA+Pegasos, suggesting that DGMs benefit a lot from max-margin learning on image classification. We also compare CMMVA with state-of-the-art results. To the best of our knowledge, there is no competitive generative models to classify digits on SVHN dataset with full labels.

Table 3: Error rates (%) on SVHN dataset.

| MODEL | ERROR RATE |
|---|---|
| *CVA+Pegasos* | 25.3 |
| *CMMVA* | 3.09 |
| *CNN* [27] | 4.9 |
| *Stochastic Pooling* [33] | 2.80 |
| *Maxout Network* [8] | 2.47 |
| *Network in Network* [17] | 2.35 |
| *DSN* [15] | 1.92 |

We further compare the generative capability of CMMVA and CVA to examine the benefits from jointly training of DGMs and max-margin classifiers. Though CVA gives a tighter lower bound of data likelihood and reconstructs data more elaborately, it fails to learn the pattern of digits in a complex scenario and could not generate meaningful images. Visualization of random samples from CVA and CMMVA is shown in Fig. 2. In this scenario, the hinge loss regularization on recognition model is useful for generating main objects to be classified in images.

### 4.4 Missing Data Imputation and Classification

Finally, we test all models on the task of missing data imputation. For MNIST, we consider two types of missing values [18]: (1) **Rand-Drop**: each pixel is missing randomly with a pre-fixed probability; and (2) **Rect**: a rectangle located at the center of the image is missing. Given the perturbed images, we uniformly initialize the missing values between 0 and 1, and then iteratively do the following steps: (1) using the recognition model to sample the hidden variables; (2) predicting the missing values to generate images; and (3) using the refined images as the input of the next round. For SVHN, we do the same procedure as in MNIST but initialize the missing values with Guassian

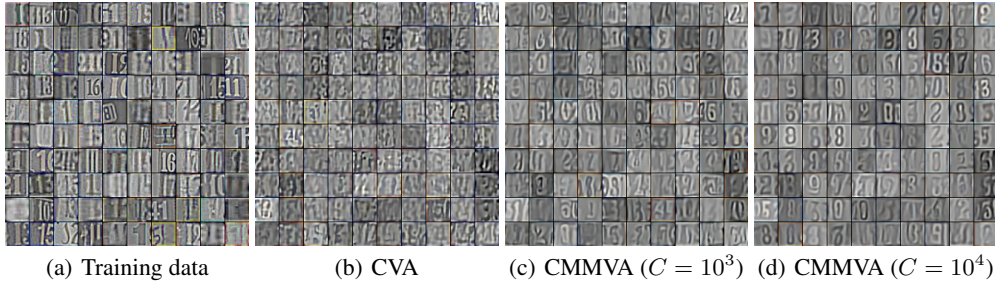

|          |          |          |          |
|:--------:|:--------:|:--------:|:--------:|
| (a) Training data | (b) CVA | (c) CMMVA ($C = 10^3$) | (d) CMMVA ($C = 10^4$) |

Figure 2: (a): training data after LCN preprocessing; (b): random samples from CVA; (c-d): random samples from CMMVA when $C = 10^3$ and $C = 10^4$ respectively.

random variables as the input distribution changes. Visualization results on MNIST and SVHN are presented in Appendix C and Appendix D respectively.

Intuitively, generative models with CNNs could be more powerful on learning patterns and high-level structures, while generative models with MLPs lean more to reconstruct the pixels in detail. This conforms to the MSE results shown in Table 4: CVA and CMMVA outperform VA and MMVA with a missing rectangle, while VA and MMVA outperform CVA and CMMVA with random missing values. Compared with the baseline, mmDGMs also make more accurate completion when large patches are missing. All of the models infer missing values for 100 iterations.

Table 4: MSE on MNIST data with missing values in the testing procedure.

| NOISE TYPE | VA | MMVA | CVA | CMMVA |
|------------|--------|--------|--------|--------|
| RAND-DROP (0.2) | 0.0109 | 0.0110 | 0.0111 | 0.0147 |
| RAND-DROP (0.4) | 0.0127 | 0.0127 | 0.0127 | 0.0161 |
| RAND-DROP (0.6) | 0.0168 | 0.0165 | 0.0175 | 0.0203 |
| RAND-DROP (0.8) | 0.0379 | 0.0358 | 0.0453 | 0.0449 |
| RECT ($6 \times 6$) | 0.0637 | 0.0645 | 0.0585 | 0.0597 |
| RECT ($8 \times 8$) | 0.0850 | 0.0841 | 0.0754 | 0.0724 |
| RECT ($10 \times 10$) | 0.1100 | 0.1079 | 0.0978 | 0.0884 |
| RECT ($12 \times 12$) | 0.1450 | 0.1342 | 0.1299 | 0.1090 |

We also compare the classification performance of CVA, CNN and CMMVA with Rect missing values in testing procedure in Appendix E. CMMVA outperforms both CVA and CNN.

Overall, mmDGMs have comparable capability of inferring missing values and prefer to learn high-level patterns instead of local details.

## 5 Conclusions

We propose max-margin deep generative models (mmDGMs), which conjoin the predictive power of max-margin principle and the generative ability of deep generative models. We develop a doubly stochastic subgradient algorithm to learn all parameters jointly and consider two types of recognition models with MLPs and CNNs respectively. In both cases, we present extensive results to demonstrate that mmDGMs can significantly improve the prediction performance of deep generative models, while retaining the strong generative ability on generating input samples as well as completing missing values. In fact, by employing CNNs in both recognition and generative models, we achieve low error rates on MNIST and SVHN datasets, which are competitive to the state-of-the-art fully discriminative networks.

**Acknowledgments**

The work was supported by the National Basic Research Program (973 Program) of China (Nos. 2013CB329403, 2012CB316301), National NSF of China (Nos. 61322308, 61332007), Tsinghua TNList Lab Big Data Initiative, and Tsinghua Initiative Scientific Research Program (Nos. 20121088071, 20141080934).

## Footnotes

[1] The source code is available at https://github.com/zhenxuan00/mmdgm.

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
