[Supplementary Material]

# A  Detailed Architectures

In all of our experiments, (C)VA and (C)MMVA share same structures and settings.

## A.1  MNIST

We set the dimension of the latent variables to 50 in all of the experiments on MNIST.

In MLP case, both the recognition and generative models employ a two-layer MLP with 500 hidden units in each layer. We illustrate the network architecture of MMVA with Gaussian hidden variables and Bernoulli visible variables in Fig. 1.

Figure 1: Network architecture of MMVA with Gaussian hidden variables and Bernoulli visible variables.

In CNN case, our convolutional network contains 5 convolution layers. There are 32 feature maps in the first two convolutional layers and 64 feature maps in the last three convolutional layers. We use filters of size 3 throughout the network except filters of size 5 in the first layer. Instead of global average pooling, a MLP with 500 hidden units is adopted at the end of the convolutional nets to obtain a tighter lower bound and better generation results. We also involve three dropout layers with keeping ratio 0.5.

## A.2  SVHN

We only consider CNN case here and the structure is similar with the CNN case on MNIST. We use 256 latent variables to capture the variation of data in pattern and scale and no MLP layer is adopted. There are 64 feature maps in the first three layers and 96 feature maps in the last two layers. We involve three dropout layers with keeping ratio 0.7.

We use a fully supervised CNN without dropout layers to initialize the recognition model of CM-MVA to speed-up the convergence of the parameters and the initial error rate is 5.03%.

# B  T-SNE Visualization Results

T-SNE embedding results of the features learned by VA and MMVA on 2D plane are shown in Fig. 2 (a) and Fig. 2 (b) respectively, using the same data points randomly sampled from the MNIST dataset. Compared to the VA's embedding, MMVA separates the images from different categories better, especially for the confusable digits such as digit "4" and "9". These results show that MMVA,

(a) VA  (b) MMVA

Figure 2: t-SNE embedding results for both (a) VA and (b) MMVA.

which benefits from the max-margin principle, learns more discriminative representations of digits than VA.

## C  Imputation Results on MNIST

(a) Original data  (b) Noisy data  (c) Results of CVA  (d) Results of CMMVA

Figure 3: (a): original test data; (b) test data with missing value; (c-d): results inferred by CVA and CMMVA respectively for 100 iterations.

The imputation results of CVA and CMMVA are shown in Fig. 3. CMMVA makes fewer mistakes and refines the images better, which accords with the MSE results as reported in the main text.

(a) Rand-Drop (0.6)  (b) Rect (12 × 12)

Figure 4: Imputation results of MMVA in two noising conditions: column 1 shows the true data; column 2 shows the perturbed data; and the remaining columns show the imputations for 20 iterations.

We visualize the inference procedure of MMVA for 20 iterations in Fig. 4. Considering both types of missing values, MMVA could infer the unknown values and refine the images in several iterations even with a large ratio of missing pixels.

| (a) Original data | (b) Noisy data | (c) Results of CVA | (d) Results of CMMVA |

Figure 5: (a): original test data; (b) test data with missing value; (c-d): results inferred by CVA and CMMVA respectively for 100 epochs.

Table 1: Error rates(%) with missing values on MNIST.

| Noise Level | CNN | CVA | CMMVA |
|---|---|---|---|
| Rect ($6 \times 6$) | 7.5 | 2.5 | 1.9 |
| Rect ($8 \times 8$) | 18.8 | 4.2 | 3.7 |
| Rect ($10 \times 10$) | 30.3 | 8.4 | 7.7 |
| Rect ($12 \times 12$) | 47.2 | 18.3 | 15.9 |

## D   Imputation Results on SVHN

We visualize the imputation results of CVA and CMMVA in Fig. 5 with Rect ($12 \times 12$) noise. In most cases, CMMVA could complete the images with missing values on this much harder SVHN dataset. In the remaining cases, CMMVA fails potentially due to the changeful digit patterns and less color contrast compared with hand-writing digits dataset. Nevertheless, CMMVA achieves comparable results with CVA on inferring missing data.

## E   Classification Results with Missing Values

We present classification results with missing values on MNIST in Table 1. CNN makes prediction on the incomplete data directly. CVA and CMMVA infer missing data for 100 iterations at first and then make prediction on the refined data. In this scenario, CMMVA outperforms both CVA and CNN, which demonstrates the advantages of our mmDGMs, which have both strong discriminative and generative capabilities.