[Reviews · NeurIPS 2015]

Submitted by Assigned_Reviewer_1

This paper introduces a max-margin criterion to the learning of a deep generative model, essentially enhancing the discriminative abilities of the latent variables z.

Unsurprisingly, this improves the classification abilities of the model, compared to the use of a simple linear classifier trained on the latent variables of the original generative model.

On the MNIST and SVHN sets, the final model is reasonably competitive with purely discriminative models, though not state-of-the-art.

However, it retains the advantage of being a generative model.

The paper is well written.

The technical sections are particularly clear, although I found the motivation section was weak and unclear in places.

Although the idea of incorporating the max-margin criterion is not highly innovative, this paper adds a large amount of useful work regarding how the optimisation should be achieved.

I am not an expert on the literature in this area, so am unable to make a strong judgement on the level of originality.

I found that the work is not clearly motivated in the early sections.

I was initially rather sceptical about whether there is any gain to using a discriminative criterion with a generative model - if you want generative behaviour, why not use a generative model; if you want discriminative behaviour, why not use a discriminative model?

However, I am prepared to accept that there are advantages.

In particular, section 4.4 is very interesting and in a sense, motivates the rest of the paper for me.

However, there is a missed opportunity to compare the classification performance against other systems in the missing data scenarios - I would find good results in this area to be a compelling reason for using the proposed model.

Simply comparing error rates against a series of discriminative models in Tables 1 and 3 aren't so interesting.

The mathematics seems correct to me, but I have not checked every equation in detail.

Guassian -> Gaussian

There are some minor inconsistencies in the references section, eg. [4] vs. [33], and [13].

Summary: This is a well-written paper with extensive, interesting theoretical analysis.

The experiments were somewhat lacking, and didn't always provide good motivation for the combination of the generative model with the discriminative max-margin criterion.

Submitted by Assigned_Reviewer_2

The work investigates how a generative model ([11] is used; others would potentially work, too) can be trained to both have good generating properties and that the latent representation of the data can be used successfully to solve a prediction task. Therefore the work assumes labeled data. The objective consists of a variational upper bound on the negative marginal likelihood of the data and a max-margin objective that measures the predictive performance of a linear classifier on the latent representation. The generative model makes use of the sampling trick presented in [11] to be able to compute gradients of the sampling procedure needed to approximate the objective. The work goes into great detail on how to optimize the objective and proposes a doubly-stochastic gradient descent method. The experiment section applies the presented model to the MNIST and Street View House Number datasets. The predicitive performance largely improves through the presented method.

This method is derived from variational principles. It is also interesting that the generated sampled seem to profit from the labels (see Fig 2.). I am curious if the varitional bound is actually tighter (so it helps the non-linear optimization problem to find a better solution) or if this is just an effect of the limited training data (in the limit the ML-estimator should find a good solution, too; modulus the approximations).

The presentation is principled, clear and approachable. The experiment section evaluates a large set of different setups: in particular the influence of the trade-off parameter in the training objective.

The derived objective bridges two interesting problems and I think this approaches a relevant problem. I would to see this work published.

Summary: The work proposes a system that jointly trains a generative model to have good generating and predicting capabilities. The training objective combines a maximum likelihood estimator for the generative process and a max-margin loss for the predictor, which is based on the inferred latent variables.

Submitted by Assigned_Reviewer_3

This paper proposes a loss function to learn deep generative models. Different previous deep generative models, this loss function has a hinge loss besides of the negative log-likelihood. The optimization problem is solved using a doubly stochastic generalization of the Pegasos algorithm. Experiments are carried out on the MNIST and SVHN datasets. The good classification and generative performances are illustrated.

The method looks novel and works well. However, there are some details which are not clear to me. 1) The network structures in the paper could be visualized using a figure. 2) In Figure 1, the quality of the generated images is pretty good. Is there any advantage or disadvantage when comparing MMVA to VA? 3) The generative performance of this method should be compared to the previous generative methods [24,12,8,2] quantitatively.

4) Is it not clear how the method optimizes the parameters in the networks? How the Doubly Stochastic Subgradient Algorithm works with back-propagation? 5) How the authors implements the MLP networks and CNN? Do you develop your algorithms based on some open-source deep learning frameworks, such as Caffe? 6) How about the time complexities of training and testing in these networks? 7) Are the max-margin deep generative models suitable for large scale image classification? For example, image classification on imagenet.
Summary: This paper proposes a loss function to learn deep generative models. This work is novel and effective, however, there are some details not clear.

Author Feedback
Author rebuttal: We thank all reviewers for their valuable comments and acknowledging our contributions. We'll improve our paper in the final version. Below, we address the comments in detail.

To R1:

Thanks for the suggestion. Indeed, we did experiments on classifying MNIST images with missing values, which weren't included due to space limit. We'll add them.

Take the rectangle noise (See Sec 4.4) in the testing procedure as an example. We compare three methods: CNN, CVA & CMMVA, which are trained on original MNIST data. A CNN with a testing error 0.43% on complete MNIST data makes prediction on the incomplete data directly. CVA and CMMVA with default settings infer missing data for 100 iteration at first and then make prediction on the refined data. The classification error rates under four noise levels (See Table 4) are CNN (7.5%, 18.8%, 30.3%, 47.2%), CVA (2.5%, 4.2%, 8.4%, 18.3%) and CMMVA (1.9%, 3.7%, 7.7%, 15.9%). Such results demonstrate the advantages of DGMs and that supervision indeed helps DGM to achieve even better accuracy when a patch is missing.

To R2:

As in Fig. 2, CVA pays same attention to the center digit and background, while CMMVA focuses on the main object to be classified. As C gets larger, images generated by CMMVA have clearer center digit but hazier background. CVA has a tight lower bound because it tries to reconstruct the whole image in detail and it gains much from the background. However, CVA fails to learn patterns in this complex scenario, while supervision helps to generate meaningful images with clear main objects.

To R3:

Q1. Network structures:
Thanks. We'll illustrate them.

Q2. Generative performance with VA & previous work:
MMVA achieves comparable qualitative and quantitative results as VA on MNIST (See Fig.1 & Table 4). For the more complex SVHN data, MMVA generates more meaningful images via the guide of supervision (See Fig.2 & lines 373-377).

For clarity and space limit, we focused on comparing with the direct competitor VA, to demonstrate how max-margin learning affects its generative capacity. But our max-margin formulation can be applied to any DGM that defines a joint distribution of (X, Z) (See lines129-130). Doing similar analysis for other DGMs is an interesting topic that we'll pursue in the future.

Q3. Optimization details:
As stated in Sec 3.2, the network parameters \phi are jointly learned as others by solving problem (6) via stochastic subgradient descent, where the subgradient over \phi is unbiasedly estimated via Eq.(7). When a multi-layer recognition network is used to parameterize the variational distribution q_\phi in Eq.(8), calculation of the derivative \nabla_\phi log q_\phi in Eq.(7) involves back-propagation. More details can also be found in [10, 11].

Q4. Implementation:
We implement MMVA based on Theano, a popular open-source framework. We'll make it clear and disclose our code.

Q5. Time complexity & Scaling up:
In training, CVA has an extra decoding procedure and a random sampling procedure, compared to CNN. Typically, the decoding procedure costs similar time as the encoding procedure, and the random sampling is fast. CMMVA has similar time complexity as CVA because the SVM part is fast to train. Namely, according to our implementation, if CNN costs time T, then CMMVA costs time 2T+t, where T>>t. In testing, all these models have similar time complexity. Thus, CMMVA can be applied to large-scale datasets, similar as CNN.

To R5:

Thanks. We'll improve the clarity of Sec-3.2. One possible reason why "likelihood regularizer" doesn't boot discriminative performance compared to pure CNN methods is that we have a sufficiently large training set to learn the deep models (both CNN and MMVA) in all the tested settings. We anticipate that with fewer labelled data (e.g., in the semi-supervised setting), the likelihood regularizer can improve. In fact, similar phenomenon was observed in [10], where combining a generative VA with the discriminative class-condition VA helps to improve the accuracy. Furthermore, we have extra results on the more challenging task of classifying images with missing values, where MMVA outperforms both VA and CNN. Please see comments to R1.

To R7:

MMVA is fundamentally different from the class-condition VA [10] in both model definition and learning objective. MMVA is novel, as agreed by other reviewers. By taking class labels as a condition to generate observations, class-condition VA needs to infer the label posterior p(y|x) for testing, which is often more expensive than our method. For example, class-condition VA takes more than ten minutes to test all images in the MNIST dataset, while MMVA takes several seconds. Furthermore, as shown in Table 1, MMVA outperforms the best model in [10], which combines VA and class-condition VA, when the recognition model is a MLP. For CNN recognition model, similar results can be expected. We'll include more analysis in the final version.